# Experimental Study at the Phase Interface of a Single-Crystal Ni-Based Superalloy Using TEM

**DOI:** 10.3390/ma15196915

**Published:** 2022-10-05

**Authors:** Hongye Zhang, Huihui Wen, Runlai Peng, Ruijun He, Miao Li, Wei Feng, Yao Zhao, Zhanwei Liu

**Affiliations:** 1School of Technology, Beijing Forestry University, Beijing 100083, China; 2School of Aerospace Engineering, Beijing Institute of Technology, Beijing 100081, China; 3School of Electrical Engineering, Hebei University of Science and Technology, Shijiazhuang 050018, China; 4AECC Beijing Institute of Aeronautical Materials, Beijing 100095, China

**Keywords:** single-crystal Ni-based superalloy, interface, strain, dislocation density, intermediate temperature brittleness

## Abstract

The single-crystal Ni-based superalloys, which have excellent mechanical properties at high temperatures, are commonly used for turbine blades in a variety of aero engines and industrial gas turbines. Focusing on the phase interface of a second-generation single-crystal Ni-based superalloy, in-situ TEM observation was conducted at room temperature and high temperatures. Intensity ratio analysis was conducted for the measurement of two-phase interface width. The improved geometric phase analysis method, where the adaptive mask selection method is introduced, was used for the measurement of the strain field near the phase interface. The strained irregular transition region is consistent with the calculated interface width using intensity ratio analysis. An intensity ratio analysis and strain measurement near the interface can corroborate and complement each other, contributing to the interface structure evaluation. Using TEM in-situ heating and Fourier transform, the change of dislocation density in the γ phase near the two-phase interface of the single-crystal Ni-based superalloy was analyzed. The dislocation density decreases first with the increase in temperature, consistent with the characteristics of metal quenching, and increases sharply at 450 °C. The correlation between the variation of dislocation density at high temperatures and the intermediate temperature brittleness was also investigated.

## 1. Introduction

Single-crystal turbine blade technology is one of the core technologies of aero engines, and almost all military and civil aircraft engine turbine blades are now made of single-crystal high-temperature alloys [1,2,3,4]. A domestic second-generation single-crystal Ni-based high-temperature superalloy, which is smelted on the basis of the CMSX-4(Ni-9.3Co-6.2Cr-6.3W-6.9Ta-5.8Al-2.9Re-1Ti-0.6Mo-0.1Hf) alloy composition, was selected as the experimental material here [5]. This material has an unusual combination of excellent physical and chemical properties, such as enhanced creep and oxidation resistance at high temperatures. Such alloys are precipitation strengthened by coherent *L*1_2_-ordered γ′ phase (Ni_3_Al precipitates) embedded in a face-centered cubic Ni-rich γ matrix, which can work stably above 1050 °C [6].

The γ/γ′ phase interface in single-crystal Ni-based high-temperature alloys has an important influence on the mechanical properties of such alloys [7,8]. One of the fundamental underlying reasons for the high-temperature microstructural stability of Ni-based superalloy is the low free energy associated with coherent γ/γ′ interfaces [3,9]. Various atomistic simulations indicate that the interface between an ordered precipitate and a disordered matrix in a Ni-based superalloy is not abrupt at the atomic scale [10,11]. With the help of modern electron microscopy, in-situ high-resolution transmission electron microscope (HR-TEM) experiments can be realized for the measurement of interface structure analysis with atomic-scale rendition [7,8,9,12,13,14]. Experimental investigation of γ′/γ interfaces in a Co-based alloy by coupling of aberration-corrected high resolution scanning transmission electron microscopy and orientation-specific atom probe tomography has established its interfaces and are both compositionally and structurally diffuse with nearly the same width [9]. It was measured by experiments that there are composition transition regions and structural transformation regions with a thickness of about 2~3 nm in the two-phase interface [15,16]. Studies have shown that the directional migration or change from the γ/γ′ interface due to the microstructure and evolution of the γ/γ′ interface leads to the directional coarsening of the γ/γ′ interface and ultimately affects the macroscopic creep properties and fracture failure [17].

At the nanoscale, the TEM experimental study is mainly to observe the material after creeping at room temperature or high temperature and the evolution of different creep times. There were few in-situ observation focused on the two-phase interface evolution. Many scholars have studied the formation mechanism and control method of different defects from the production process, the strengthening mechanism of elements, the evolution of alloy phases, the structure of each interface in a topologically close-packed phase, and the interaction between dislocation and phase interface. In terms of the interaction between dislocation and phase interface, scholars [18,19] have proposed different models, including dislocation glide and climbing, where molecular dynamics and first principles are always used for calculation.

Single-crystal Ni-based superalloys are always a research hotspot for their wide applications. For this study, in-situ TEM observation was conducted at room temperature and high temperatures, focusing on the phase interface of a second-generation single-crystal Ni-based superalloy. An intensity ratio analysis was conducted for the measurement of two-phase interface width. An improved geometric phase analysis method was used to measure the strain field near the phase interface. The strained irregular transition region is consistent with the calculated interface width using intensity ratio analysis. After in-situ heating, the change of dislocation density in the γ phase near the two-phase interface of the tested alloy was analyzed. The correlation between the variation of dislocation density at high temperatures and the intermediate temperature brittleness was also investigated. The present study and any related work at high spatial resolution and high temperature may play a major role in understanding the overall mechanical behavior of these superalloys.

## 2. Experimental and Materials

The tested material is a second-generation single-crystal Ni-based high-temperature superalloy whose composition is similar to the CMSX-4 alloy. Table 1 reports its chemical composition with ‘other’ indicating trace elements, which were measured by a fluorescence spectrophotometer (PANalytical Axios-mAx, PANalytical B.V., Almelo, Holand) [16]. The experimentally used rod-shaped single crystal alloy in this paper was prepared by melting the parent alloy in a vacuum induction furnace and then by a directional solidification technique with a crystal orientation of <001>, followed by multiple-step heat treatment, including solution treatment at 1300 °C/2 h (air cooling, AC) + 1080 °C/4 h (AC) + 1120 °C/4 h (AC) + 900 °C/4 h (AC). TEM samples were prepared by a focused ion beam electron beam (FIB-EB) cross-beam system. A Pt layer with a thickness of about 2 um was deposited in the area of interest. The main purpose is to reduce the damage and implantation of Ga+ ions on the sample during the thinning process. Finally, the thickness of the thin area is less than 100 nm. Details of the technique can be found in the literature [20]. In-situ observations near the γ/γ′ phase interface were carried out with a TEM (Tecnai G2 F20, FEI, OR, USA) at an accelerating voltage of 200 kV. EDS mapping was carried out and mapping images of Ni, Al, Ta, Cr, Co, Re, Mo, and W are shown in Figure 1.

A FEI NanoEx-i/v device, a single-tilt TEM specimen heating, and biasing holder, was used to realize TEM in-situ heating. The in-situ observation was realized by tracking two natural markers near the γ/γ′ interface. The HRTEM images were captured at different temperatures, with a temperature interval of 50 °C from 50 °C to 600 °C, cooling to 400 °C, and room temperature (RT). The TEM sample was observed at different temperature stages, with ending times of 395 s, 815 s, 2030 s, 2560 s, 3030 s, 3460 s, 4410 s, 5550 s, 6550 s, 7550 s, 8470 s, 9730 s, 9920 s, 11,450 s corresponding to 50 °C, 100 °C, 150 °C, 200 °C, 250 °C, 300 °C, 350 °C, 400 °C, 450 °C, 500 °C, 550 °C, 600 °C, 400 °C, and back to RT. During constant temperature observations, the temperature fluctuations were plus or minus 0.2 °C. The raw HRTEM images were properly filtered to enhance their contrast before quantitative calculation.

## 3. Methods

In this work, the intensity ratio analysis method [8,16] is used for calculating the interface width of the γ/γ′ interface. In the HRTEM images, a region with a clear atomic arrangement is selected to calculate the atomic column intensity of the γ and γ′ phases, respectively. Due to the Ni atom in the γ phase and the alternate appearance of heavy and light atoms in the γ′ phase, the intensity curve of the γ phase tends to be stable, while the intensity curve of the γ′ phase has the phenomenon of alternating peaks and valleys (zigzag distribution). A transition region would appear between the two phases and can be used to determine the position of the γ/γ′ interface. The amplitude of zigzag distribution would become much less from γ′ phase to γ phase crossing the transition zone. Thus, the interface width can be identified.

Fourier transformation (FT) has had a wide and profound impact, with specific applications in many fields such as physics, optics, and materials sciences since its birth. By combining FT and lattice image processing, the deformation information of the lattice structure can be calculated [21]. The geometric phase analysis (GPA) is used for quantitative analysis of the strain field near the γ/γ′ interface. The GPA method, introduced by Hÿtch [22,23], is successfully applied in the displacement/strain field measurement of crystal structures at the nanoscale. Improved by Liu [24], the so-called subset-GPA (S-GPA) was introduced, where the windowed Fourier transformation is added in the transforming process. The core theoretical formula of S-GPA [24] is as follows.

The 2-dimensional windowed Fourier transform (2D-WFT) is:(1)$$Q(\mu ,\upsilon ,\xi ,\eta )=\int_{-\infty }^{\infty }\int_{-\infty }^{\infty }q(x,y)g(x-\mu ,y-\upsilon )\exp\{-i\xi x-i\eta y\}dxdy$$
where Q(μ,υ,ξ,η) represents the windowed Fourier spectrum; ξ and η represent the frequency components in the *x* and *y* directions, respectively; g is a window function representing the reciprocal lattice vector of the lattice; (μ,υ) is the coordinate of the center of the window and the target window changes with different pairs of μ and υ.

The 2D inverse WFT is:(2)$$\hat{q}(x,y)=\frac{1}{4\pi^{2}}\int_{-\infty }^{\infty }\int_{-\infty }^{\infty }\int_{-\infty }^{\infty }\int_{-\infty }^{\infty }\bar{Q}(\mu ,\upsilon ,\xi ,\eta )g(x-\mu ,y-\upsilon )\times \exp\{i\xi x+i\eta y\}d\xi d\eta d\mu d\upsilon$$
with
(3)$$\bar{Q}(\mu ,\upsilon ,\xi ,\eta )=\left\{ \begin{array}{l} Q(\mu ,\upsilon ,\xi ,\eta ),\left|Q(\mu ,\upsilon ,\xi ,\eta )\right|/max(\left|Q(\mu ,\upsilon ,\xi ,\eta )\right|)\;\;\geq Thr\\ 0,\;\;\;\;\;\;\;\;\;\;\;\;\;\;\;\;\;\left|Q(\mu ,\upsilon ,\xi ,\eta )\right|/max(\left|Q(\mu ,\upsilon ,\xi ,\eta )\right|)\;\;<Thr \end{array}\right.$$
where $\bar{Q}(\mu ,\upsilon ,\xi ,\eta )$ represents the filtered frequency spectrum;  |Q(μ,υ,ξ,η)| represents the power of Q(μ,υ,ξ,η); the mostly used Gaussian window function g is divided by $\sqrt{\pi \sigma_{x}\sigma_{y}}$ for normalization; Thr is the preset threshold.

The wrapped phase can be obtained through the simple arctangent function $\hat{q}(x,y)$. Using the phase unwrapping technique, the original phase field and the phase difference can be obtained. The displacement/strain can be calculated through the following matrix form equation:(4)$$\left(\begin{array}{cc} \varepsilon_{xx} & \varepsilon_{xy}\\ \varepsilon_{yx} & \varepsilon_{yy} \end{array}\right)=\left(\begin{array}{cc} \frac{\partial u(x)}{\partial x} & \frac{\partial u(x)}{\partial y}\\ \frac{\partial u(y)}{\partial x} & \frac{\partial u(y)}{\partial y} \end{array}\right)=-\frac{1}{2\pi }{\left(\begin{array}{cc} g_{1x} & g_{1y}\\ g_{2x} & g_{2y} \end{array}\right)}^{-1}\left(\begin{array}{cc} \frac{\partial P_{g_{1}}}{\partial x} & \frac{\partial P_{g_{1}}}{\partial y}\\ \frac{\partial P_{g_{2}}}{\partial x} & \frac{\partial P_{g_{2}}}{\partial y} \end{array}\right)$$
where the subscripts *x* and *y* represent the *x* and *y* directions, u(x) and u(y) are the displacements in the *x* and *y* directions, $P_{g}$ is the phase in the image, $\varepsilon_{xx}$ and $\varepsilon_{yy}$ are the direct strains, $\varepsilon_{xy}$ and $\varepsilon_{yx}$ are the shear strains, respectively.

GPA requires the selection of diffraction spots, which involves the choice of diffraction spot size and shape as well as different filtering methods that apply to the specific form of the mask plate. In the Fourier filtering process, the diameter or width of the mask plate will determine the effective spatial resolution and frequency resolution, which in turn determine the form of the strain field fluctuations. By reducing the width of the data smoothing region (reducing the size of the filter window), a smoother strain field distribution can be obtained, resulting in a loss of spatial information and spectral resolution. To obtain reliable strain information, the relationship between selected spatial resolution and frequency resolution must be compromised. In the literature [24], the influence of the mask size on the calculation results of Global-GPA (G-GPA) and S-GPA calculations has been specifically compared by simulations.

In the G-GPA method, the window shape is usually circular or square, and the displacement and strain fields can be obtained with high accuracy if the window size is appropriate. In the calculation of bidirectional fringes, the window size and shape are the same, which is suitable for the calculation of bidirectional fringes with similar general frequencies. However, in some special cases, the conventional window shape cannot meet the measurement requirements if the fringe has a specific deformation or a large difference in frequency or signal strength in two directions. The results obtained by simply selecting the circular or square window form and adjusting the window size cannot correctly reflect the deformation. For the areas with small diffraction spots, the filtering is not complete, resulting in errors. Therefore, we further improve the selection of window shapes and develop a window form with an adjustable shape, namely, the adaptive mask selection method. This method can adjust the window shape in different directions according to needs and can be adjusted to an ellipse or arbitrary shape.

As shown in Figure 2, Figure 2a is a constructed bi-directional orthogonal lattice image. The vertical direction contains large deformations, and the frequency of the lattice in the horizontal and vertical directions is inconsistent. The inset denotes the Fourier transform spectrum of Figure 2a. Figure 2c,g are the central regions of the diffraction spectrum after amplification. Figure 2d,e denote the fringe image of region 1 and region 2 after IFFT in Figure 2c, separately. The fringe image of region 1 of Figure 2g after IFFT is the same as in Figure 2d. Region 2 of Figure 2g is the modified selected diffraction spot shape, of which the region can also be arbitrary. Figure 2h denotes the fringe image of region 2 of Figure 2g after IFFT. Figure 2j,k,l are the true value of V-field displacement, the calculated value of the conventional circular window, and the calculated value of the elliptical correction window, respectively. The fringe image obtained after IFFT is more consistent with the deformation of the original lattice, and the fringe continuity of the corresponding order is better after the improved adaptive mask selection method is adopted. It is difficult to obtain good results using conventional window forms, as shown in the red and yellow dashed areas in Figure 2f,i. After using the improved shape-adjustable window, the deformation obtained from subsequent calculations will be more consistent with the true deformation of the sample, which is conducive to improving the accuracy of the measurement results under the appropriate working conditions.

The window function is to reduce the influence of leakage in the filtering process, which is inevitable in practical applications. However, applying a window function before the FFT can effectively reduce the influence of leakage. The cost is that more fluctuation distribution will appear in the deformation field obtained in the subsequent calculation. The specific forms of filtering windows include rectangular windows, Lorentz windows, Gaussian windows, sinusoidal windows, sine windows, Hanning windows, and Hamming windows. The previous studies on window functions have been more comprehensive and systematic [25,26,27,28]. The most common Gaussian window is used here.

Here, a typical common distortion field is used by simulation to analyze and compare the measurement results of G-GPA and improved S-GPA method.
(5)$$q\left(x,y\right)=\frac{255}{4}\left(\sin\left\{\frac{2\pi }{d}x+\phi \left[\frac{3\left(x-256\right)}{256},\frac{3\left(y-256\right)}{256}\right]\right\}-\sin\left(\frac{2\pi }{d}y\right)+2\right)$$
with
(6)$$\begin{array}{l} \phi \left(x,y\right)=3{\left(1-x\right)}^{2}\cdot \exp\left\{-x^{2}-{\left(y+1\right)}^{2}\right\}-10\left(\frac{x}{5}-x^{3}-y^{5}\right)\\ \;\;\;\;\;\;\;\;\;\;\;\;\;\cdot \exp\left\{-x^{2}-{\left(y+1\right)}^{2}\right\}-\frac{1}{3}\exp\left\{-{\left(x+1\right)}^{2}-y^{2}\right\} \end{array}$$

Figure 3a is a constructed lattice image containing deformation by using Equations (5) and (6). The true strain value *ε_xx_* is shown in Figure 3b, with a maximum absolute strain value of 0.24. Figure 3c,d show the results calculated by the improved S-GPA and G-GPA, respectively. The window parameters used in the S-GPA are the suggested values from the literature [29]. The window size is set to 10, and the size of the diffraction spot chosen in the G-GPA is *g*/3.

The evaluation parameter of calculation reliability (EPCR) [24] is used to quantitatively investigate the performance of the S-GPA method and to make a comparison between the two methods: S-GPA and G-GPA. The EPCR is set to a certain absolute value of relative error as a percentage of all calculation areas. Figure 3 and Figure 4 demonstrate that the calculation error of S-GPA is lower than that of G-GPA within the calculation range of the GPA method, where the points with strain values of less than 500 με are removed from the statistics. The percentage of calculable areas for the S-GPA and G-GPA methods were calculated to be 91.17% and 83.80% when the absolute relative error values are less than 10%, and 81.29% and 68.85% when the absolute relative error values are less than 5%.

In fact, due to the nature of spectrum processing, there is always a certain volatility or ripple effect in the obtained strain field, which is also reflected in the error field in Figure 4. However, the analysis reveals that the measurement error of the S-GPA method developed in this chapter is consistently better than that of the G-GPA method thanks to the WFT algorithm used in S-GPA, which performs WFT block by block in the image and can treat the local area as uniformly deformed regions, which allows for more accurate extraction of local frequencies and the non-uniformly deformed fields to be measured with higher accuracy.

## 4. Results and Discussion

### 4.1. Phase Interface Width and Strain Distribution

Figure 5a,b show the high-resolution images taken near the phase interface of the single-crystal nickel-based high-temperature superalloy when the electron beam incidence direction is [001]. The determined two-phase interface width is shown in Figure 5c,d. Figure 5c shows the intensity sum obtained by superimposing the gray value along the [020] direction. The intensity profile changes significantly from the γ′ phase to the γ phase, and there is a transition region. Figure 5d shows a clear order-disorder transition zone at the γ/γ′ interface (A–B), where long-range order decreases roughly over 12–14 atomic layers in [020] direction (approximately 2 to 2.4 nm). The same transition zone has also been marked at the corresponding location in Figure 5c. Similarly, a compositional transition zone (A–C in Figure 5c, measuring 4.31 nm), from the lower intensity columns on the γ′ side to the higher intensity columns on the γ side can now be defined. In this case, the calculated atomic distribution fits better with the findings in the literature [8,16]. The irregular effects in the pink dashed areas in Figure 5c,d are caused by the irregular distribution of atomic gray value in the pink dashed area in Figure 5b, where the irregular atomic distribution may be influenced by multiple factors, such as imaging conditions, tiny local stresses, etc. The subsequent strain calculations in Figure 6 also show an irregular strain distribution.

To obtain the real strain field near the two-phase interface, the strain field was calculated using the S-GPA method [24] by selecting a reference region in each of the two phases (the yellow dashed area in Figure 5a). The calculating procedure is similar to that used in the measurement of real strain and stress in heterogeneous structures [30,31]. The calculated strain fields are shown in Figure 6, where the strains in the [020] and [$\bar{2}00$] directions were obtained. There is a relatively large non-uniform deformation inside the γ phase. On the whole, the strain field distribution near the two-phase interface is not uniform (mainly in the [$\bar{2}00$] direction). The reason is that the lattice parameters of the two phases are different due to the inhomogeneous distribution of different elements. In both the γ′ and γ phases and the transition region, there are many different elements with different atomic sizes, different interaction forces, and small differences in the relative positions between the atoms, which cause the inhomogeneous distribution of the strain field. In the strain fields near the two-phase interface, the strain field distribution in the [$\bar{2}00$] direction shows more clearly that there is an anomaly in the strain distribution near the phase interface. It can be expected that in the strain field near the two phases, due to the different lattice constants of the two phases, the γ′ phase is under tension near the interface, while the γ phase is under pressure near the interface, so that the region within the dashed lines A to C can better reflect this regularity.

Figure 7 shows the profile after averaging the strain along [020] in the area corresponding to the pink dashed lines in Figure 5a and Figure 6b. The strain profile in the area of the dashed line AC reflects the tensile properties of the γ′ phase and the compressive properties of the γ phase at the two-phase interface when the tensile strain distribution on the left side of the dashed line C is not considered. When comparing Figure 5 and Figure 6, the calculated transition area of the strain field’s phase interface matches the area contained in the dashed AC line in Figure 5c of the intensity ratio analysis better, indicating that the real strain field analysis is more representative of the deformation and structural morphology near the phase interface. The HRTEM-based intensity ratio analysis can better characterize the width of the phase interface, but there is inevitably the influence of human selection error in the full characterization of the transition region near the phase interface.

Assuming that the area within the dashed AC line is the phase interface region, the phase interface width was about 4.31 nm. In our previous study [16], the measured interface width is about 2 nm. The phase interface width of the tested alloy is not a constant value and varies from site to site. The analysis [16] assumes that the position of the phase interface in the width direction is identical in the observed area. While the analysis in Figure 5a directly uses all the grayscale data in the [020] direction, the distribution of the intensity and sum of the dashed AC segment in Figure 5c would result from calculation errors if the phase interface is not completely horizontally distributed. The tensile and compressive strains near the dashed AB section in Figure 6b are also not perfectly regular, further indicating that the phase interface is not perfectly distributed but has some fluctuations. In short, the strains vary greatly during the transition from γ′ to γ phase at the phase interface, and the width of the strain irregularity transition region is the same as the width of the phase interface calculated by the intensity ratio analysis. The determination of the two-phase interface based on HRTEM images and the calculation of the strain field near the two-phase interface can corroborate and complement each other.

### 4.2. Variation of Dislocation Density and Intermediate Temperature Brittleness

Here, the effects of dislocations are further analyzed at high temperatures, specifically the changes and their effects on the number of dislocations during in-situ observation. One of the evaluation indicators of dislocations in material science is the dislocation density, which is the total length of dislocation lines in a unit volume. It can be expressed as ρ=L/V, where *L* is the total length of dislocation lines and *V* is the crystal volume. Dislocation density can also be expressed as the number of dislocation lines per unit area
(7)$$\rho =\frac{Nl}{Sl}=\frac{N}{S}$$
where *S* is the crystal area, l is the dislocation line length, *N* is the number of dislocation outcrops in the *S* surface, and the unit of dislocation density is generally 1/cm^2^.

In this study, the observation method based on HRTEM images [32] is used to determine the number of dislocations and estimate the dislocation density of the specimens according to Equation (7) by combining high-resolution images and Fourier transform images. The lattice images at different temperatures were obtained by FFT of the high-resolution images and IFFT of the same size as the diffraction spots in the [020] direction, as shown in Figure 8.

In Figure 8, the dislocations at different temperatures in the γ phase near the two-phase interface are marked, and each dislocation is obtained by transforming the lattice image in the [020] direction and combining it with the original high-resolution image. Since the direct inversion of the lattice image by the FFT method has errors when the image quality is poor, it needs to be combined with the original high-resolution image for judgment. The white ‘T’ shape denotes a definite dislocation, while the pink ‘T’ shape is not completely sure to be a dislocation, and it is considered a suspected dislocation. The white dashed ellipse is the area where the original image is found to be unclear, and the lattice images of the upper and lower parts of the area are needed to determine whether there are any dislocations and to identify them. The pink dashed ellipse area is where the original image is very unclear, and the presence of dislocations cannot be accurately determined with the help of the upper and lower parts of the visible lattice image, which is a region of greater uncertainty. In particular, it should be noted that there is no more visible lattice information around the edges, and the dislocations presented at individual edges were not statistically accounted for. Among them, the dark purple circular dashed area at 200 °C is a clear lattice fringe image not captured in the original image, where the overall grayscale value is small and black. In addition, in combination with the surrounding visible lattice image, it can be judged that there is no dislocation here. In addition, the dislocation distribution also shows that the dislocations have a tendency to cross the phase interface into the γ′ phase.

By counting the number of dislocations determined at each temperature, i.e., the dislocations marked with white ‘T’ shapes in Figure 8, and bringing in Equation (7), the variation of dislocation density with temperature can be calculated, as shown in Figure 9. It can be found that the dislocation density of the tested superalloy tends to decrease, then increase significantly, and then decrease again with the change in temperature.

Intermediate-temperature brittleness refers to the metal and alloy in the range of about 0.5–0.8 melting point where the elongation or section shrinkage is significantly reduced. The fracture is along the crystal fracture and is accompanied by impurity elements biased toward the grain boundaries. Here, the dislocation density increased significantly at 450 °C and then decreased. Dislocation density has an important effect on the intermediate-temperature brittleness of an alloy. High dislocation density makes it easy for dislocations to cross each other during dislocation movement, forming cut steps and causing dislocation entanglement, then causing obstacles to dislocation movement and making it difficult to continue plastic deformation, finally increasing the strength of the material. This contributes to the intermediate-temperature brittleness of the material. In fact, the physical properties of materials change dramatically at the nanoscale. For example, the melting point of nanomaterials decreases dramatically compared to macroscopic bulk materials [33,34]. The melting point of bulk Ni is 1458 °C, while the melting point of Ni when the particle diameter is approximately 20 nm is 700 °C [35]. In conclusion, the Ni element occupies the largest proportion in the tested superalloy, and its nanoscopic properties will greatly affect the properties of the tested alloy at the nanoscale. The melting point of the alloy at the nanoscale will be reduced to about half of that of the bulk material, and considering the influence of refractory elements, the melting point of the tested alloy at the nanoscale would be about 700–800 °C. This manner, the maximum temperature in this study is very close to the service temperature at the macroscopic level of the alloy. Compared to the existing studies [36,37] relating to single-crystal Ni-based alloys, which exhibit intermediate temperature brittleness around 700 °C, the intermediate-temperature brittleness of the tested alloy here at the nanoscale is most likely around 350–400 °C. In the present study, there is a significant dislocation density increase at 450 °C, which can correspond to its macroscopic intermediate-temperature brittleness to some extent. The dislocation density increase, therefore, makes an important contribution to the intermediate-temperature brittleness phenomenon of single-crystal Ni-based high-temperature superalloys.

## 5. Conclusions

The main findings in the current work can be summarized as follows:(1)The S-GPA method has been enhanced. The mask selection form in the GPA method is introduced, and the adaptive mask selection method is used to solve the problem of low accuracy of GPA measurement under non-uniform large deformation.(2)Intensity ratio analysis was conducted for the interface width evaluation of a single-crystal Ni-based superalloy. The strain field containing the two phases was obtained by using S-GPA in the two phases by selecting a reference region, respectively. The width of the calculated two-phase interface is basically consistent with the width of the strain-irregular transition region. The determination of the two-phase interface based on HRTEM images and the calculation of the strain field near the two-phase interface can corroborate and complement each other.(3)TEM in-situ heating was conducted. The change of dislocation density in the γ phase near the two-phase interface of the single-crystal Ni-based superalloy was analyzed. The dislocation density decreases first with the increase in temperature, which is consistent with the characteristics of metal quenching, and increases sharply at 450 °C. This is closely related to the intermediate temperature brittleness of the alloy, where dislocations cause a non-equilibrium bias of solute atoms at medium temperatures, which increases the yield strength of the alloy and causes the intermediate temperature brittleness of the alloy.

## Figures and Tables

**Figure 1 materials-15-06915-f001:**
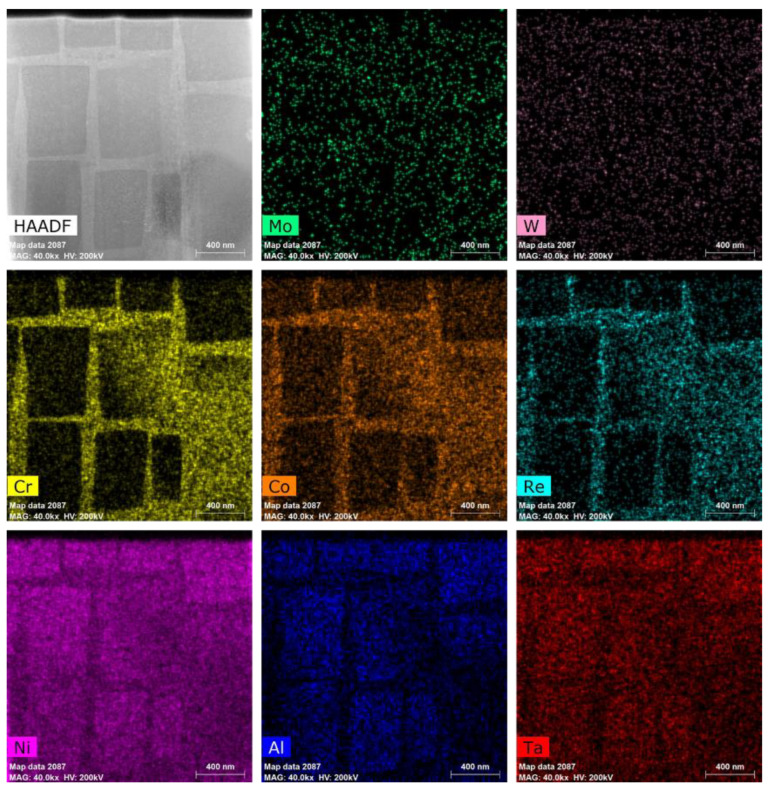
HADDF-EDS mapping of the single-crystal Ni-based superalloy.

**Figure 2 materials-15-06915-f002:**
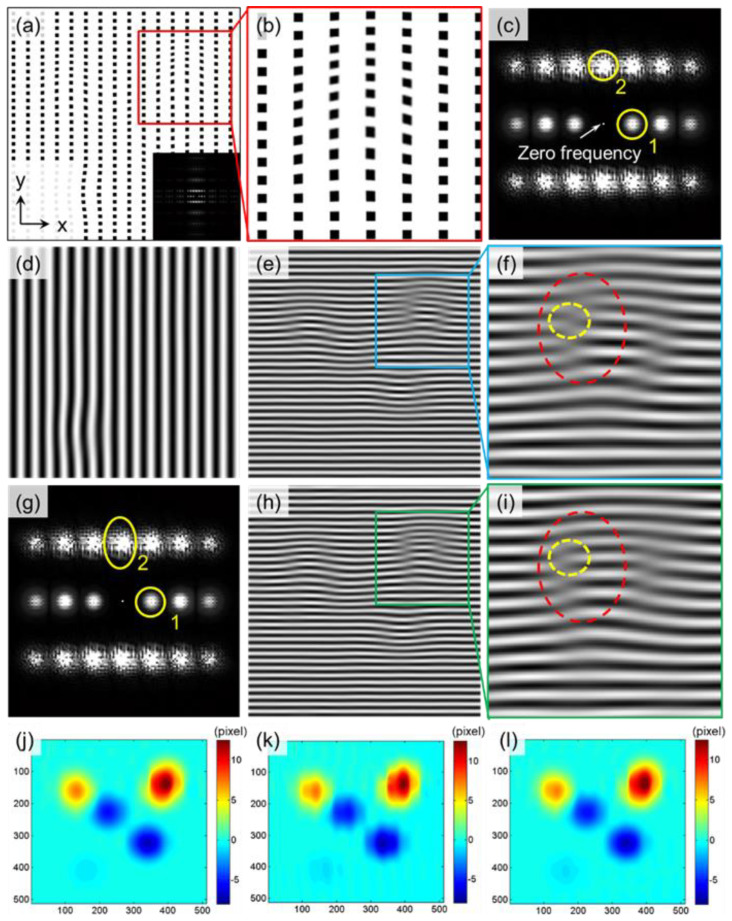
(**a**,**b**) Constructed lattice image; (**c**) Diffraction spectrum; (**d**) Fringe image of region 1 after IFFT in (**c**); (**e**,**f**) Fringe image representing region 2 after IFFT in (**c**); (**g**) Diffraction spectrum; (**h**,**i**) Fringe image representing region 2 after IFFT in (**g**); V−field displacement; (**j**) True value; (**k**) Calculated value using conventional circular window; (**l**) Calculated value using elliptical correction window.

**Figure 3 materials-15-06915-f003:**
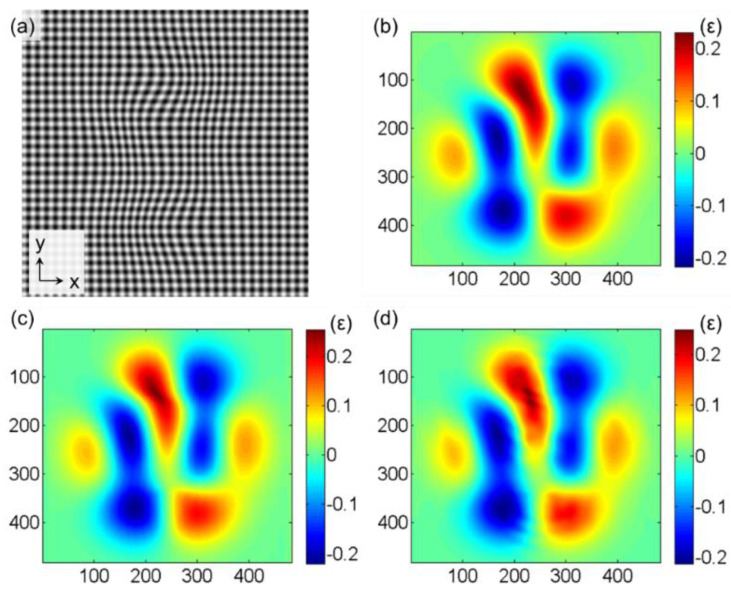
(**a**) Constructed lattice image containing deformation; Comparison results of *ε_xx_* (**b**) by presetting; (**c**) by improved S−GPA; and (**d**) by G−GPA.

**Figure 4 materials-15-06915-f004:**
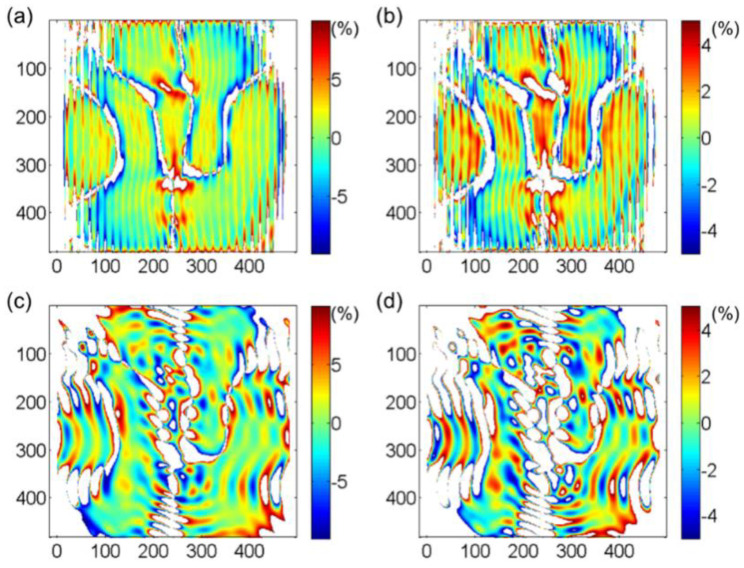
Strain error field calculated by S−GPA when the EPCR is (**a**) 10% and (**b**) 5%; Strain error field calculated by G−GPA when the EPCR is (**c**) 10% and (**d**) 5%.

**Figure 5 materials-15-06915-f005:**
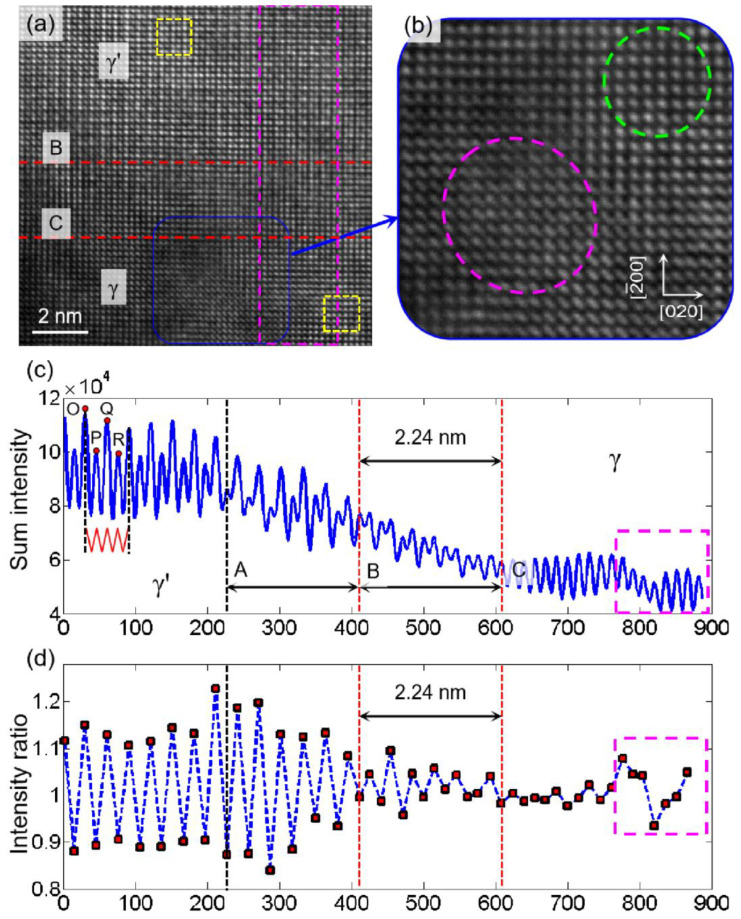
Intensity ratio analysis. (**a**,**b**) Filtered HRTEM image at the two-phase interface; (**c**) Sum intensity profiles across the [020] direction of the pink dashed box showing the transition from ordered γ′ to disordered γ; and (**d**) Intensity ratio of each atomic column to its adjacent column on the right.

**Figure 6 materials-15-06915-f006:**
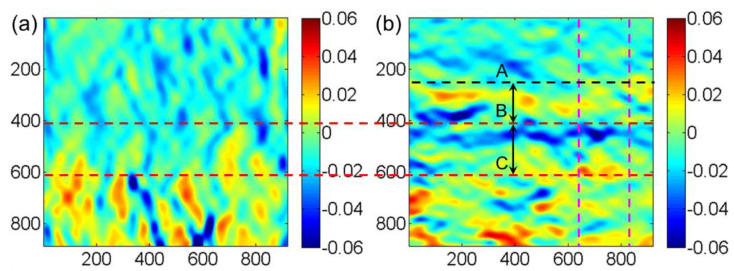
Strain distribution near the two-phase interface at room temperature: (**a**) [020] direction and (**b**) [$\bar{2}00$] direction.

**Figure 7 materials-15-06915-f007:**
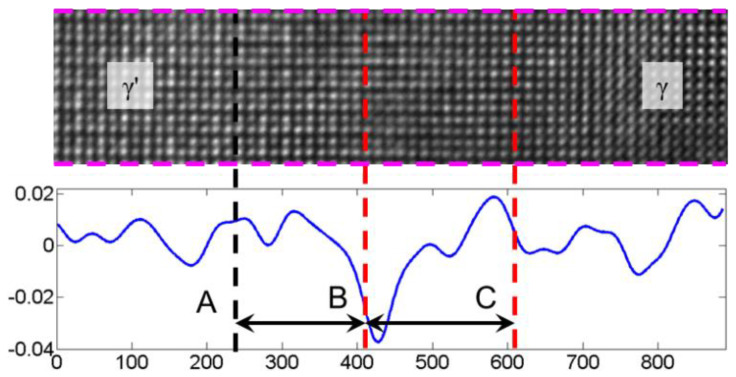
Average strain distribution profile along the [020] direction near the two-phase interface.

**Figure 8 materials-15-06915-f008:**
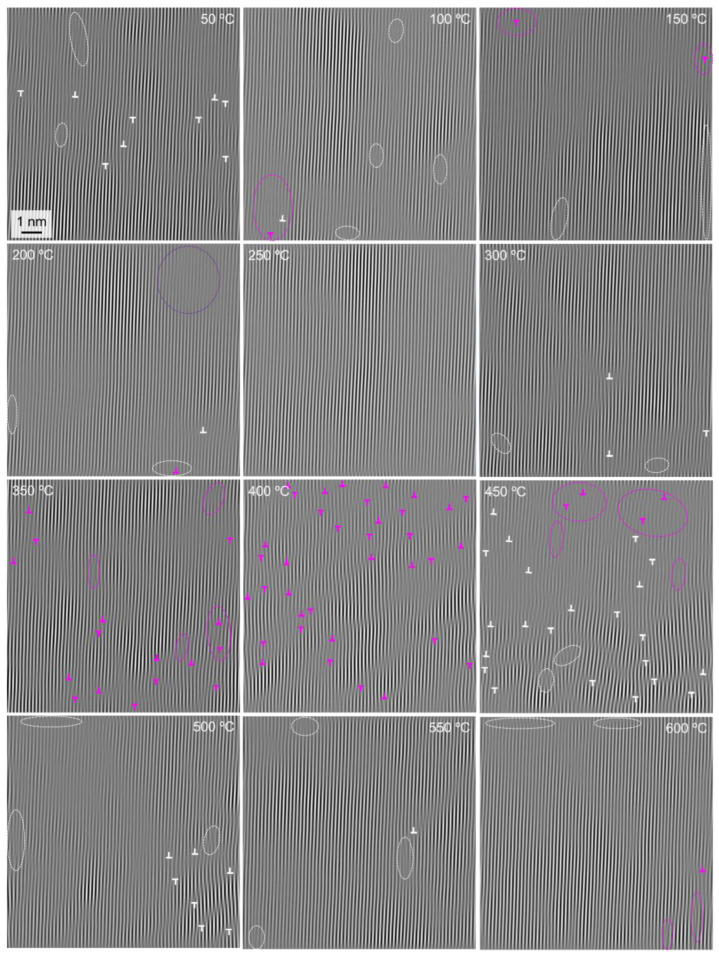
Changes in the number of dislocations in the γ phase near the two-phase interface during heating.

**Figure 9 materials-15-06915-f009:**
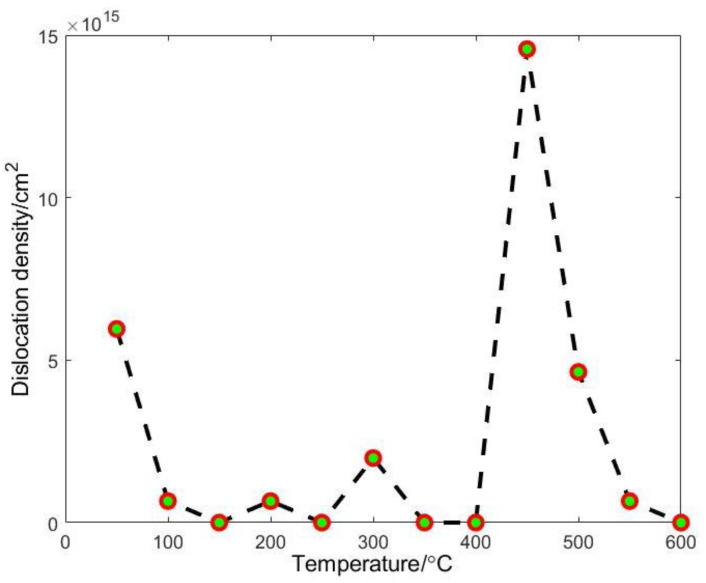
Variation of dislocation density with temperature.

**Table 1 materials-15-06915-t001:** Composition of the single-crystal Ni-based superalloy as measured by a fluorescence spectrophotometer (wt./%).

Element	Cr	Co	W	Al	Ta	Mo	Re	Hf	Other	Ni
Content	7.05	7.76	4.90	6.17	6.61	1.50	3.01	0.13	<0.1	Bal.

## Data Availability

Not applicable.

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
