# Peer review of "Experimental Study at the Phase Interface of a Single-Crystal Ni-Based Superalloy Using TEM"

_materials, 2022, doi:10.3390/ma15196915_

Round 1

Reviewer 1 Report

The submitted manuscript is of high quality, and I have remarks mainly about the editing issues:

1. Line 33 - Authors should provide the chemical composition of the CMSX-4 alloy they mention.

2.  Line 83 - a reference is missing.

3.  Lines 173, 209, 210. 211. 222, 223, 242, 245, etc. - Errors in references that need to be corrected.

4. Figures 6 and 7 - there is a lack of description of the axis.

5. Figure 8 - there is a lack of marker.

Author Response

We would like to thank you for acknowledging our effort. The following is our response to the suggestions raised in your comments.

We have carefully revised the manuscript according to your and the other reviewers’ insightful suggestions. We sincerely hope the revised manuscript will be finally acceptable to be published on Materials. The revised places are marked in red in the revised manuscript.

Reviewer 2 Report

Dear Authors!

The manuscript is well-organized and presented. It contains the improvement of the calculation methods, and the results may be applied to other PH-hardened alloys. The explanation and proof of mid-temperature brittleness are well-argued. Please, check minor spelling errors in the manuscript

Regards,

Author Response

Thanks for your kind reminder and acknowledging our effort.

We have amended the mistakes your raised in your comments, line 34, 71, 78, 83, 89, 123, 130, 173, 209, 210, 211, 222, 242, 245, 255, 257, 259, 265, 287-289, 292-293, 295, 307, and 311.

In line 324, the corresponding symbols have been revised into italics.

We are sorry that the editing error upset your reading experience. We hope the revised version will please your eyes as well as the possible general audience.

According to your insightful comments and suggestions, we have amended the manuscript with more care. By the way, we think the revised manuscript has shown a vast improvement. We sincerely hope the revised manuscript will be accepted to be published on Materials.

Reviewer 3 Report

In lines 30-32, the authors have mentioned that all military and civilian turbines are made of single crystal superalloy. While this issue should be clarified. There are different blades in a turbine, each of which has its own unique type and production method.

The sentence leading to line 36 needs a reference. for example “10.1007/s12540-019-00465-2”

In line 39, what is the difference between γ/γ׳  phase and γ/γ׳  eutectic?

Line 83 of this sentence is stated “indicating Y, B et al., which is measured by a fluorescence spectrophotometer”. What do you mean? If it is a quote, why is there no reference?

The authors have used the word single crystal in the title of the manuscript, then in the experiments section (line 86), they mentioned that the superalloy was prepared by directional solidification???

Heat treatment at 1300 oC is a very high temperature (line 87). According to what standard did you choose this temperature?

In most cases, there is no documentation for the sentences used. For example, there has been a lot of talk about dislocations. But a clear TEM image of the dislocation along the γ׳ is not included in the manuscript.

Author Response

We have carefully revised the manuscript according to your and the other reviewers’ insightful suggestions. We sincerely hope the revised manuscript will be finally acceptable to be published on Materials. The revised places are marked in red in the revised manuscript.

Round 2

Reviewer 3 Report

The manuscript is well improved and deserves to be published in the Materials. Questions were answered well. But I am still not convinced by the two answers below.

1- It is true that the directional solidification structure has similarities with a single crystal. But their differences are also evident. Therefore, there is no match between the title section and the experimental section. Either call both directional solidification or both single crystal.

2- I am still not convinced why the heat treatment temperature of 1300 was chosen. It is true that you have obtained the sample from the supplier and they have given you these specifications, but there must be a scientific reason. The temperature of 1300 is a high temperature and there is a possibility of deformation of the sample, formation of harmful phases of TCP, and other harmful things unless there is a special reason. I would be interested to know your answer to this. Good luck.

Author Response

Response to Reviewer 3 Comments

The manuscript is well improved and deserves to be published in the Materials. Questions were answered well. But I am still not convinced by the two answers below.

Response:

We would like to thank you for acknowledging our effort. The following is our response to the suggestions raised in your comments.

1- It is true that the directional solidification structure has similarities with a single crystal. But their differences are also evident. Therefore, there is no match between the title section and the experimental section. Either call both directional solidification or both single crystal.

Response:

Thanks for your kind suggestion.

We contacted our supplier. It should be the “grain selector method”. Honestly, not being an expert in the field of manufacturing single crystal alloys we are not familiar with the specific technical details enough. And there is misunderstanding of the important concepts. Thanks for your kind suggestion. We will and should learn more knowledge relating to this field.

2- I am still not convinced why the heat treatment temperature of 1300 was chosen. It is true that you have obtained the sample from the supplier and they have given you these specifications, but there must be a scientific reason. The temperature of 1300 is a high temperature and there is a possibility of deformation of the sample, formation of harmful phases of TCP, and other harmful things unless there is a special reason. I would be interested to know your answer to this. Good luck.

Response:

Thanks for your question.

As mentioned in the response to question 1, we contacted our supplier to reconfirm the process parameters. Fortunately, we did not record incorrect data. You are right that 1300 °C is a very high temperature. Unfortunately, we feel sorry that we don't have the capacity to discuss the scientific reasons in depth now. To our knowledge, there are heat treatment parameters in published articles that are higher than 1300 °C. Heat treatment is to find a balance between macro and micro performance differences, including strength, ductility, fracture toughness, fatigue resistance, creep and oxidation resistance, and so on. The mechanism of all these factors is important. We think all the mechanisms are not very clear and there are many development directions. And this is why the supperalloy is always a hotspot in the research field, where we are also one of the researchers. As you can see, we should learn and research more to understand this field in depth.

Thanks for acknowledging our effort and encouraging us to pursue our work. As an expert in this field, your comments and questions are very pertinent and professional. We will continue to conduct research in this area, and we hope we can obtain some meaningful findings. Thanks again for all your help.
